# Circulating CD8+ T Cell Subsets in Primary Sjögren’s Syndrome

**DOI:** 10.3390/biomedicines11102778

**Published:** 2023-10-13

**Authors:** Igor Kudryavtsev, Stanislava Benevolenskaya, Maria Serebriakova, Irina Grigor’yeva, Evgeniy Kuvardin, Artem Rubinstein, Alexey Golovkin, Olga Kalinina, Ekaterina Zaikova, Sergey Lapin, Alexey Maslyanskiy

**Affiliations:** 1Federal State Budgetary Institution “Almazov National Medical Research Centre” of the Ministry of Health of the Russian Federation, St. Petersburg 197341, Russia; 2Federal State Budgetary Educational Institution of Higher Education Academician I.P. Pavlov First St. Petersburg State Medical University of the Ministry of Healthcare of Russian Federation, St. Petersburg 197022, Russia

**Keywords:** Sjögren’s syndrome, T cell subsets, CD8+ T cells

## Abstract

Currently, multiple studies have indicated that CD8+ T lymphocytes play a role in causing damage to the exocrine glands through acinar injury in primary Sjögren’s syndrome (pSS). The aim of this research was to assess the imbalance of circulating CD8+ T cell subsets. We analyzed blood samples from 34 pSS patients and 34 healthy individuals as controls. We used flow cytometry to enumerate CD8+ T cell maturation stages, using as markers CD62L, CD28, CD27, CD4, CD8, CD3, CD45RA and CD45. For immunophenotyping of ‘polarized’ CD8+ T cell subsets, we used the following monoclonal antibodies: CXCR5, CCR6, CXCR3 and CCR4. The findings revealed that both the relative and absolute numbers of ‘naïve’ CD8+ T cells were higher in pSS patients compared to the healthy volunteers. Conversely, the proportions of effector memory CD8+ T cells were notably lower. Furthermore, our data suggested that among patients with pSS, the levels of cytotoxic Tc1 CD8+ T cells were reduced, while the frequencies of regulatory cytokine-producing Tc2 and Tc17 CD8+ T cells were significantly elevated. Simultaneously, the Tc1 cell subsets displayed a negative correlation with immunoglobulin G, rheumatoid factor, the Schirmer test and unstimulated saliva flow. On the other hand, the Tc2 cell subsets exhibited a positive correlation with these parameters. In summary, our study indicated that immune dysfunction within CD8+ T cells, including alterations in Tc1 cells, plays a significant role in the development of pSS.

## 1. Introduction

Primary Sjögren’s syndrome (pSS) is an autoimmune epithelitis, which is characterized by a lymphocytic infiltration of the lacrimal and salivary glands, leading to the loss of appropriate tear and saliva production. The clinical hallmarks of pSS are keratoconjunctivitis sicca, xerostomia and parotid gland swelling [1]. Extraglandular manifestations of pSS develop in one-third of patients, leading to immune complex deposition and chronic inflammation in various organs such as skin, joints, lungs, liver, kidneys, blood vessels and even the nervous system. Glandular lesions in patients with pSS are characterized by the massive infiltration of inflammatory cells, which include CD3+ T cells, CD19+ B cells, natural killer cells, dendritic cells and macrophages [2]. In 18–59% of patients, lymphocytic infiltration of the lacrimal and salivary glands may become like secondary lymphoid organs, forming ectopic germinal centers. Extensive inflammation of the glands leads to progressive acinar epithelial cell atrophy and fibrosis [3].

Hyperactivity of B cells is thought to play an important role in the development of pSS. The current body of evidence strongly suggests this B cell hyperactivity is mediated by T cells. T cells may also be involved in a loss of self-tolerance, and they secrete many proinflammatory cytokines associated with local inflammation in pSS, including IFNγ, IL-17 and IL-21 [4].

Previously, there was an idea of predominance of activated CD4+ helper T cells (Th) in the infiltrates, and the role of CD8+ cytotoxic T cells (Tcyt) in disease pathogenesis has been unexplored [5]. Currently, the role of Tcyt is strongly revised. The CD8+ T cells are a complex group of lymphocytes, which are known to play a critical role in particular in autoimmune disease. CD8+ T cells provide their effector function in three ways: by releasing granules filled with cytotoxic molecules, through Fas–FasL interactions, and by producing cytokines. Overactivity or abnormal proliferation of CD8+ T lymphocytes can be detected in the peripheral circulation and target tissues of pSS patients [6].

A pathogenic role of CD8+ T cells in lacrimal gland autoimmunity was shown in fundamental work with nonobese diabetic (NOD) mouse, which is a well-characterized model of pSS. The transfer of purified CD8 T cells isolated from the cervical lymph nodes of NOD mice into NOD-SCID (severe combined immunodeficiency) recipients resulted in inflammation of the lacrimal glands. These data demonstrate a pathogenic role of CD8 T cells in lacrimal gland autoimmunity [7].

In the study by Mingueneau M et al., patients with pSS had increased CD8+ HLA-DR+ cells in peripheral blood. Likewise, salivary gland biopsy revealed abundant infiltration of CD8+ T lymphocytes, which mostly had the CD8+HLA-DR+ phenotype. HLA-DR+ CD8+ T cells in salivary gland biopsy and peripheral blood positively correlated with the activity of the disease, showing their important role in pSS [8].

Fujihara T et al. showed that CD8+ but not CD4+ T lymphocytes were located around the acinar epithelial cells, and the ascensive effector molecules of cytotoxic CD8+ T lymphocytes such as granzyme B, perforin, IFNγ and tumor necrosis factor-α (TNFα) can be detected. These results indicate the presence of a population of activated CD8+ T lymphocytes with cytotoxicity that are competently able to lead to the death or apoptosis of the cells [9].

In the study Kaneko N et al., granzyme-A positive CD8+ T cells were the most prominent T cells in the labial salivary glands from untreated patients with pSS. It was noted that acinar cells and ductal cells undergo apoptosis, and CD8+ T cells contribute to apoptotic cell death, using both granule exocytosis and Fas signaling to destroy target cells [10].

However, Zhang X et al. investigated the regulatory function of CD8+ cells in pSS pathogenesis using a murine desiccating stress model that resembles pSS [11].

Here we report a study of CD8+ cells peripheral blood subsets of patients with pSS. We used flow cytometry to immunophenotype peripheral blood from 34 patients with pSS and 34 control subjects. Then we analyzed the relationship between CD8+ T cell subsets, clinical manifestations of the disease and disease activity.

## 2. Materials and Methods

All patients with pSS (n = 34) met the 2016 ACR-EULAR Classification Criteria for pSS [12]. Control subjects (n = 34) were healthy volunteers with no signs of autoimmune disease. The groups were comparable in gender and age. ESSDAI (EULAR Sjögren’s Syndrome Disease Activity Index) and ESSPRI (EULAR Sjögren’s Syndrome Patient Reported Index) were calculated to assess the activity of the disease [13,14]. The Schirmer test was performed without the use of local anesthetics using special test strips (Contacare, Mumbai, India). Unstimulated sialometry was measured according to generally accepted recommendations [15]. Minor labial salivary gland biopsies were obtained from all pSS patients to confirm the diagnosis. The clinical characteristics of pSS patients in this study are presented in the Table 1.

All participants provided signed informed consent. All the samples were collected for scientific research in accordance with the Helsinki Declaration and were given the green light by the local ethics committee (protocol No. 03-20 16.03.2020).

### 2.1. Laboratory and Pathological Determination

The levels of total rheumatoid factor (RF), immunoglobulin G (IgG), C-reactive protein (CRP), C3 and C4 complement were investigated using an automatic Abbott analyzer. Erythrocyte sedimentation rate was investigated by Westergren method.

The method of indirect immunofluorescence on a commercial substrate (Euroimmun AG, Lübeck, Germany) was used to detect the antinuclear factor (ANF). The line-blot method with commercial reagent kits (Euroimmun AG, Germany) was used to identify the spectrum of antinuclear autoantibodies, such as double-stranded DNA (Anti-DNA Ab), anti-SSA-antibodies (SSA Ab), anti-SS-B-antibodies (SSB Ab), anti-Ro-52antibodies (Ro-52 Ab) and anti-centromere B-antibodies (CENP B).

### 2.2. Sample Collection

Peripheral blood samples were taken from the patients before starting the treatment. About five milliliters of peripheral blood was gathered from each patient using anticoagulant tubes containing K3EDTA. The collected blood samples were promptly processed. The immunophenotyping of CD8+ T cell subsets was carried out within a few hours (less than 6 h) after collecting the blood. To measure cytokine levels, plasma samples without cells were obtained by spinning the whole blood at 300× *g* for 7 min at +4 °C. This was followed by transferring it to clean 1.5 mL tubes and spinning again at 300× *g* for 15 min at +4 °C to remove leftover platelets and other blood cells. Finally, each plasma sample was divided into smaller portions and stored at −80 °C until needed.

### 2.3. Immunophenotyping of Circulating CD8+ T Cell Subset Maturation Stages and ‘Polarized’ CD8+ T Cell Subsets

For immunophenotyping of CD8+ T cell subset maturation stages, whole peripheral blood samples in an amount of 200 μL each were stained with the following monoclonal antibodies: ECD-labeled mouse anti-human CD62L, PC5.5-labeled mouse anti-humanCD28, PC7-labeled mouse anti-human CD27, APC-labeled mouse anti-humanCD4, APC-AF700-labeled mouse anti-human CD8, APC-AF750-labeled mouse anti-human CD3, Pacific Blue-labeled mouse anti-human CD45RA and Krome Orange-labeled mouse anti-human CD45 (all antibodies were manufactured by Beckman Coulter, Indianapolis, IN, USA, and were used according to the manufacturer’s recommendations). Next, all samples were incubated at room temperature in the dark for 15 min followed by red blood cell lysis for 15 min in the dark with 2 mL of VersaLyse Lysing Solution (Beckman Coulter, Inc., Indianapolis, IN, USA) supplemented with 50  μL of IOTest 3 Fixative Solution (Beckman Coulter, Inc., Indianapolis, IN, USA). Finally, 200 μL of Flow-Count Fluorospheres (Beckman Coulter, Indianapolis, IN, USA) was added, and sample acquisition was performed using a 3/10 Navios flow cytometer (Beckman Coulter, Indianapolis, IN, USA). At least 20,000 CD8+ T cells were analyzed in each sample. The gating strategy for CD8+ T cell subset maturation stages was described previously in detail [16].

For immunophenotyping of ‘polarized’ CD8+ T cell subsets, whole peripheral blood samples in an amount of 200 μL each were stained with the following monoclonal antibodies: FITC-labeled mouse anti-human CD45RA (Beckman Coulter, Indianapolis, IN, USA), PE-labeled mouse anti-human CD62L (Beckman Coulter, Indianapolis, IN, USA), PerCP/Cy5.5-labeled mouse anti-human CXCR5 (CD185, BioLegend, Inc., San Diego, CA, USA), PE/Cy7-labeled mouse anti-human CCR6 (BioLegend, Inc., San Diego, CA, USA), APC-labeled mouse anti-human CXCR3 (CD183, BioLegend, Inc., San Diego, CA, USA), APC-Alexa Fluor 750-labeled mouse anti-human CD3 (Beckman Coulter, Indianapolis, IN, USA), Pacific-Blue-labeled mouse anti-human CD8 (BioLegend, Inc., San Diego, CA, USA) and Brilliant Violet 510-labeled mouse anti-human CCR4 (CD194, BioLegend, Inc., San Diego, CA, USA). All antibodies were used according to the manufacturer’s recommendations). Blood samples were treated with antibodies at room temperature in the dark for 15 min. Following this, erythrocytes were broken down by adding 1 mL of VersaLyse Lysing Solution (Beckman Coulter, Inc., Indianapolis, IN, USA) along with 25 µL of IOTest 3 Fixative Solution (Beckman Coulter, Inc., Indianapolis, IN, USA) in the dark at room temperature for another 15 min. Subsequently, all samples were washed twice (at 330× *g* for 8 min) using sterile PBS supplemented with 2% fetal calf serum (FCS) from Sigma-Aldrich Co., Saint Louis, MO, USA. The samples were then suspended in 500 µL of fresh PBS containing 2% neutral formalin (Sigma-Aldrich Co., Saint Louis, MO, USA) and were analyzed using flow cytometry. A minimum of 20,000 CD3+CD8+ Th cells were gathered from each sample. The methodology used to identify the primary ‘polarized’ CD8+ T cell subsets was previously explained in comprehensive detail [16].

### 2.4. Cytokine and Chemokine Measurement

We evaluated the levels of 47 cytokines, chemokines and growth factors in plasma samples. This assessment was carried out using a multiplex analysis on fluorescently labeled magnetic microsphere beads. We used the MILLIPLEX^®^ MAP Human Cytokine/Chemokine/Growth Factor Panel A (HCYTA-60K-PX48, MilliporeSigma, Burlington, MA, USA), following the manufacturer’s instructions. The analysis was performed on a Luminex MAGPIX^®^ (RUO) Instrument (Luminex, Austin, TX, USA), as previously described in detail [17].

### 2.5. Statistical Analysis

The flow cytometry data were examined utilizing Kaluza software v2.3 (Beckman Coulter, Indianapolis, IN, USA). The multiplex analysis data were all generated using xPONENT v.4.3 software and were assessed using Milliplex Analyst 5.1 Flex software (Luminex, Austin, TX, USA). For statistical analysis, Statistica 7.0 (StatSoft, Tulsa, OK, USA) and GraphPad Prism 8 (GraphPad Software Inc., San Diego, CA, USA) software packages were employed.

To assess normality, the Pearson’s chi-squared test was applied. Flow cytometry data were presented as cell percentages, and the absolute count of CD8+ T cell subsets was calculated utilizing Flow-Count Fluorospheres (Beckman Coulter, Indianapolis, IN, USA), which were used to determine absolute counts on the flow cytometer and were presented as the number of cells per 1 μL of whole peripheral blood. All data from the multiplex analysis were reported as concentrations of cytokines, chemokines and growth factors in pg/mL. The data were represented as medians and interquartile ranges (Me: Q25; Q75). Intergroup variances were evaluated through a nonparametric Mann–Whitney U test with a significance threshold of *p* < 0.05.

## 3. Results

### 3.1. Main Peripheral Blood CD3+ T Cell Subsets in Patients with Sjögren’s Syndrome

Preliminarily, peripheral blood samples from 34 patients with pSS and 34 samples from healthy controls (HC) were analyzed for the main T cell subsets. Our results indicated these two groups had similar percentages and absolute numbers of CD3+ cells, as shown in Figure 1. We found that the frequency of CD3+ cells in patients with pSS was 80.13% (71.13; 83.23), and very similar results were obtained for HC (76.77% (72.58; 80.44) with *p* = 0.211), while the absolute numbers of CD3+ cells were 1204 cells/1 μL (909; 1811) vs. 1083 cells/1 μL (967; 1521), respectively, with *p* = 0.492. Next, we compared the frequencies of CD4+ T cells and found that in pSS group, the relative number of Th cells was 46.15% (37.34; 54.63) vs. 47.83% (43.27; 52.42) in healthy controls (*p* = 0.404). We also found no significant differences in CD4+ T cell concentrations between the groups (650 cells/1 μL (475; 1167) vs. 670 cells/1 μL (580; 855), respectively, with *p* = 0.650). Finally, we analyzed the frequencies of CD8+ T cells, and, as shown in Figure 1, no significant differences were detected in relative and absolute numbers between the groups (25.21% (19.86; 30.14) vs. 24.26% (19.26; 28.13) with *p* = 0.418, and 459 cells/1 μL (305; 628) vs. 350 cells/1 μL (267; 487) with *p* = 0.093, respectively).

Next, we consider Spearman correlations between the levels of CD3+, CD4+ and CD8+ T cells and disease parameters of pSS.The relative number of Tcyt had significant direct correlation with IgG level (r = 0.408, *p* = 0.032).

### 3.2. Alterations in CD8+ T Cell Maturation Subsets in Patients with Sjögren’s Syndrome

Next, using multicolor flow cytometry, we classified peripheral blood CD8+ T cell subsets by CD45RA and CD62L antigens coexpression. Our combination of CD45RA, and CD62L allowed identification of ‘naïve’ CD8+ T cells (coexpressing CD45RA and CD62L), central memory CD8+ T cells (CM, with CD45RA−CD62L+ phenotype), effector memory CD8+ T cells (EM, CD45RA and CD62L double-negative cells), and terminally differentiated CD45RA-positive effector memory CD8+ T cells (TEMRA, with CD45RA+CD62L− phenotype) within total CD8+ T cell subsets [18]. We assessed the percentage and relative numbers of four main CD8+ T cell maturation subsets, and data obtained are summarized in Figure 2. Interestingly, we found that the relative and absolute numbers of ‘naïve’ CD8+ T cells were increased in patients with pSS if compared to HC (34.02% (18.86; 42.97) vs. 21.49% (14.94; 28.47) with *p* = 0.002, and 147 cells/1 μL (62; 226) vs. 69 cells/1 μL (51; 122) with *p* = 0.002, respectively). We also noticed that the percentages of EM CD8+ T cells was significantly decreased in patients with pSS (19.25% (15.93; 25.52) vs. 27.73% (22.96; 38.21) with *p* < 0.001), as shown in Figure 2.

Conducting a correlation test between peripheral blood maturation CD8+ T cell subsets and disease parameters, we found, that ‘naïve’ Tcyt was inversely correlated with CRP level (r = −0.382, *p* = 0.034), EM Tcyt was directly correlated with IgG (r = 0.486, *p* = 0.009) and CM Tcyt was directly correlated with SSA Ab (r = 0.361, *p* = 0.043).

### 3.3. Imbalance in EM and TEMRA CD8+ T Cell Subsets in Patients with Sjögren’s Syndrome

EM and TEMRA had many characteristics of effector cells, were able to migrate to inflamed tissues and could be closely associated with the functions of CD8+ T cells. Preliminarily, we investigated the expression of CD27 and CD28 costimulatory molecules on the cell surface of the CD45RA–CD62L– CD8+A T cell subset and defined the four distinct effector memory subsets, as described previously by Romero et al. [19]. Thus, CD27 and CD28 coexpression resulted in identifying CD27+CD28+EM1 cells, CD27+CD28−EM2 cells, double-negative CD27−CD28−EM3 cells as well as the CD27−CD28+EM4 cell subset (Figure 3). Our data revealed a significantly higher increase in both relative and absolute numbers of EM3 and decreased EM1 CD8+ T cells in patients with pSS compared to healthy controls (45.19% (34.91; 61.30) vs. 18.43% (10.96; 31.24) with *p* < 0.001 and 39 cells/1 μL (19; 56) vs. 16 cells/1 μL (10; 38) with *p* = 0.009 for EM3 subsets, respectively, and 30.86% (20.04; 45.41) vs. 61.67% (51.19; 67.97) with *p* < 0.001 and 24 cells/1 μL (16; 39) vs. 57 cells/1 μL (40; 91) with *p* < 0.001 for EM1 cells, respectively).

Finally, based on CD27 and CD28 expression, the TEMRA CD8+ T cells were subdivide into pre-effector type 1 cells (pE1, with CD27+CD28+ phenotype), pre-effector type 2 cells (pE2, with CD27+CD28− phenotype) and effector cells (Eff, with CD27−CD28− phenotype), as suggested by Rufer et al. [20] and Koch et al. [21]. We found that the relative and absolute numbers of two immature TEMRA CD8+ T cell subsets—pE1 and pE2—were significantly decreased in peripheral blood samples from patients with pSS syndrome if compared to healthy controls (5.13% (2.39; 11.27) vs. 16.88% (8.36; 23.74) and 6 cells/1 μL (3; 12) vs. 18 cells/1 μL (10; 32) with *p* < 0.001 in both cases for pE1 cells, and 9.91% (5.39; 18.07) vs. 18.19% (12.61; 22.76) with *p* = 0.001 and 10 cells/1 μL (5; 22) vs. 18 cells/1 μL (13; 28) with *p* = 0.004 for pE2 cells, respectively). Moreover, the frequency of mature CD27−CD28− effector TEMRA CD8+ T cells was increased in patients with pSS vs. healthy controls (81.52% (72.83; 87.68) vs. 64.42% (52.78; 77.55), *p* < 0.001). The results was shown on the Figure 4.

Thereafter, we assessed the correlations of the levels of maturation TEMRA CD8+ T cell subsets and disease parameters of pSS. There were significant positive correlations for relative number of EM3 cells and IgG and SSA and SSB Ab levels (r = 0.491, *p* = 0.008, r = 0.350, *p* = 0.049, and r = 0.425, *p* = 0.015, respectively). Also, EM2 CD8+ T cells frequency was directly correlated with erythrocyte sedimentation rate (r = 0.477, *p* = 0.016), while EM1 cells level was inversely correlated with CRP (r = −0.564, *p* < 0.001). As well, there were significant negative correlations for relative pE1 Tcyt and CRP (r = −0.396, *p* = 0.028).

Thus, our data suggest that EM and TEMRA CD8+ T cell subsets were enriched with most mature cells with effector functions as well as effector CD8+ T cells, which may function in the pathogenesis of pSS.

### 3.4. Imbalance in Peripheral Blood CD8+ T Cells ‘Polarization’ in Patients with Sjögren’s Syndrome

To evaluate relevant ‘polarized’ CD8+ T cell subsets, we studied CXCR3 and CCR6 coexpression, as it was proposed earlier [22,23]. Thus, we identified four major CD8+ T cell subsets—Tc1 (CCR6−CXCR3+), Tc2 (CCR6−CXCR3−), Tc17 (CCR6+CXCR3−) and double-positive Tc17.1 (CCR6+CXCR3+)—within the total CD8+ T cell subsets (Figure 5). We noticed that the relative numbers of CXCR3-expressing CD8+ T cells—Tc1 and Tc17.1—were decreased in patients with pSS (59.66% (50.66; 67.47) vs. 71.38% (64.85; 77.76) with *p* < 0.001 and 2.56% (1.58; 3.39) vs. 3.99% (2.71; 7.18) with *p* = 0.003, respectively), while the frequencies of Tc2 and Tc17 CD8+ T cells were elevated in patients with pSS if compared to healthy controls (30.53% (25.49; 41.10) vs. 19.26% (25.49; 41.10) with *p* < 0.001 and 1.42% (0.84; 2.86) vs. 0.82% (0.56; 1.50) with *p* = 0.013, respectively). Similarly, the absolute numbers of CD8+ T cells Tc2 and Tc17 CD8+ T cells were elevated in patients with pSS vs. healthy control group (149 cells/1 μL (76; 218) vs. 71 cells/1 μL (59; 113) with *p* < 0.001 and 7 cells/1 μL (4; 11) vs. 3 cells/1 μL (2; 6) with *p* = 0.006), while the levels of Tc17.1 was decreased (11 cells/1 μL (7; 18) vs. 14 cells/1 μL (11; 23) with *p* = 0.036).

Thus, our data indicated that in circulating blood from patients with pSS, the level of cytotoxic Tc1 CD8+ T cells that were able to kill infected cells and to secrete effector cytokines (IFNγ and TNFα) was decreased, while the frequencies of regulatory cytokine-producing Tc2 and Tc17 CD8+ T cells were significantly elevated.

To further explore the relationship between the peripheral blood CD8+ T cell ‘polarization’ and disease activity, correlation with the disease activity index and markers of disease activity were tested. There were significant negative correlations for relative number of Tc1 and IgG level and level of RF (r = −0.398, *p* = 0.036 and r = −0.608, *p* = 0.040, respectively). On the contrary, relative number of Tc2 was directly correlated with RF (r = 0.634, *p* = 0.030). Moreover, ‘naïve’ Tc1 cells, EM Tc1 and TEMRA Tc1 were inversely correlated with RF (r = −0.504, *p* = 0.023, r = −0.613, *p* = 0.004 and r = −0.753, *p* < 0.001, respectively), while ‘naïve’ Tc2, EM Tc2 and TEMRA Tc2 cells were directly correlated with it (r = 0.512, *p* = 0.021, r = 0.717, *p* < 0.001 and r = 0.771, *p* < 0.001, respectively). In addition, IgG level was inversely correlated with EM Tc1 and TEMRA Tc1 (r = −0.396, *p* = 0.037 and r = −0.463, *p* = 0.013, respectively), while it was directly correlated with EM Tc2 and TEMRA Tc2 (r = 0.393, *p* = 0.038 and r = 0.459, *p* = 0.014, respectively).

Next, Tc17 and Tc17.1 were inversely correlated with the CRP level (r = −0.430, *p* = 0.016 and r = −0.375, *p* = 0.038, respectively) and SSB antibody (r = −0.482, *p* = 0.005 and r = −0.621, *p* < 0.001, respectively). Also, CM Tc17.1, EM Tc17, EM Tc17.1 and TEMRA Tc17 were inversely correlated with the CRP level (r = −0.440, *p* = 0.013, r = −0.431, *p* = 0.015, r = −0.369, *p* = 0.004 and r = −0.364, *p* = 0.044, respectively). SSB Ab was inversely correlated with ‘naïve’ Tc17, CM Tc17.1, EM Tc17, EM Tc17.1, TEMRA Tc17. and TEMRA Tc17.1 (r = −0.408, *p* = 0.020; r = −0.495, *p* = 0.004; r = −0.350, *p* = 0.049; r = −0.492, *p* = 0.004; r = −0.406, *p* = 0.020 and r = −0.525, *p* = 0.002, respectively). SSA Ab was inversely correlated with CM Tc17.1 (r = −0.396, *p* = 0.025). Anti-DNA Abs level was directly correlated with naïve Tc17.1 and CM Tc17 (r = 0.599, *p* = 0.007 and r = 0.633, *p* = 0.007, respectively) but were inversely correlated with CM Tc1 and EM Tc1 (r = −0.499, *p* = 0.029 and r = −0.468, *p* = 0.043, respectively).

Given the correlation between CD8+ T cell subsets and markers of disease activity, we asked if the former were linked to organ involvement. Loss of appropriate tear and saliva production are the clinical hallmarks of pSS. We found significant negative correlation for relative number of Tc1, naïve Tc1, EM Tc1, TEMRA Tc1 and Shirmer test (r = −0.483, *p* = 0.02; r = −0.54, *p* = 0.008; r = −0.501, *p* = 0.015 and r = −0.518, *p* = 0.011). Also, there was significant positive correlation of Tc2, naïve Tc2, EM Tc2 and Shirmer test (r = 0.484, *p* = 0.02; r = 0.509, *p* = 0.013 and r = 0.468, *p* = 0.024). Moreover, unstimulated saliva production flow was inversely correlated with relative numbers of Tc1, naïve Tc1, CM Tc1, EM Tc1, EM2 Tc, EM3 Tc and Eff Tc (r = −0.709, *p* = 0.0003; r = −0.597, *p* = 0.004; r = −0.718, *p* = 0.0002; r = −0.774, *p* = 0.00004; r = −0.543, *p* = 0.011; r = −0.457, *p* = 0.037 and r = −0.546, *p* = 0.01), while it was directly correlated with Tc17, Tc17.1, naïve Tc2, CM Tc2, CM Tc17, EM Tc17, TEMRA Tc17, TEMRA Tc17.1 and pE1 Tcyt (r = 0.598, *p* = 0.004; r = 0.448, *p* = 0.042; r = 0.605, *p* = 0.004; r = 0.531, *p* = 0.013; r = 0.653, *p* = 0.001; r = 0.652, *p* = 0.001; r = 0.490, *p* = 0.024; r = 0.495, *p* = 0.022 and r = 0.535, *p* = 0.012).

As for other domains, relative amount of CD3+ T cells was inversely correlated with lung damage in the form of fibrosing alveolitis (R = −0.583, *p* ≤ 0.05). Moreover, relative value of Tc1 was directly correlated with it, while relative value of Tc2 and Tcyt TEMRA had inverse correlations (R = 0.648, *p* ≤ 0.05, R = −0.648, *p* ≤ 0.05 and R = −0.583, *p* ≤ 0.05, respectively). There was significant negative correlation of EM2 Tcyt and CM Tc1 with peripheral nervous system involvement (R = −0.692, *p* ≤ 0.05 and R = −0.780, *p* ≤ 0.05, respectively). Also, pE2 Tcyt was inversely correlated with joint damage (R = −0.534, *p* ≤ 0.05).

### 3.5. Blood Level of ‘Polarized’ CD8+ T Cell Subsets and Correlations with Cytokines and Chemokines

We determined the correlations of peripheral blood ‘polarized’ CD8+ T cell subsets with main serum chemokines and cytokines. Therefore, we used multiplex analysis to measure the serum levels of 47 cytokines/chemokines/growth factors in plasma samples of 20 patients with pSS.

Preliminarily, we analyzed possible correlations between IP-10 (CXCL10), MCP-1 (CCL2), MCP-3 (CCL7), MDC (CCL22), MIG (CXCL9), MIP-1a (CCL3), MIP-1b (CCL4), IL-8 (CXCL8), eotaxin (CCL11), fractalkine (CX3CL1) and main ‘polarized’ CD8+ T cell subsets (Figure 6). We noticed that serum levels of two CXCR3 ligands—CXCRL9 and CXCRL10—negatively correlated with total Tc17.1 and ‘naïve’ Tc17.1 frequencies. Similarly, the negative correlations were detected between total and effector memory Tc17, as well as total, central memory, effector memory and TEMRA Tc17.1 cells and serum levels of CCL2, which acts during physiological immune defense and chronic inflammation to activate the migration of myeloid and lymphoid cells [24]. Next, the frequencies of circulating Tc1 CD8+ T cell subsets of different maturation states negatively correlated with serum levels of CCL7 andCCL3, which play an important part in inflammatory events by attracting macrophages and monocytes to further amplify inflammatory processes [25], and in regulating lymph node homing of dendritic cell subsets as well as inducing antigen-specific CD4+ and CD8+ T cell responses [26]. Furthermore, CCL3 serum levels positively correlated with the increased frequencies of circulating ‘naïve’ Tc2 cells, as well as effector memory Tc2, Tc17 and Tc17.1 CD8+ T cell subsets, which were unregulated in patients with pSS, as shown previously.

Next, we analyzed possible correlations between main proinflammatory and effector cytokines and ‘polarized’ CD8+ T cell subsets in patients with pSS (Figure 7). Interestingly, we found no significant correlations between serum levels of main proinflammatory cytokines—IL-1a, IL-6, IL-18 and TNFα—and ‘polarized’ CD8+ T cell subsets in patients with pSS. However, we noticed positive correlations between IL-1b and several Tc2 subsets (including total Tc2 cells and CM, EM and TEMRA Tc2 cells); conversely, we found negative correlations between IL-1b levels and Tc1 subset frequencies, including total Tc1 cells and CM, EM and TEMRA Tc1 cells. Similarly, ‘non naïve’ Tc1 CD8+ T cells negatively correlated with such cytokines as IL-12p70, IL-13 andIL-17A serum levels, which regulate type 2 and type 3 inflammation [27].

## 4. Discussion

In our studies described here, we focused on maturation and ‘polarization’ subsets of peripheral blood CD8+ T cells in patients with pSS. Primarily, we found no differences in related and absolute numbers of CD3+CD8+ cells between the pSS group and healthy controls. Several research groups also noticed that patients with pSS and controls groups had no differences in CD8+ T cell frequency [28,29] or absolute count [8]. However, Sudzius et al. showed that the absolute counts of CD8+ T cells were significantly lower in pSS patients in comparison to controls [30]. Conversely, Li et al. noticed that pSS patients had an increased level of circulating peripheral blood CD8+ T cells that positively correlated with multiple disease parameters as well as the serum levels of the main effector cytokine IFNγ [31]. Furthermore, these CD8+ T cells were highly activated and expressed elevated levels of CD38 and HLA-DR on their cell membranes.

Next, we found that the relative and absolute numbers of ‘naïve’ CD8+ T cells were increased in patients with pSS, pointing to impaired differentiation and maturation of CD8+ T cells in the thymus. Interestingly, in patients with pSS, thymus enlargement was identified in approximately 22% of cases [32]. Interestingly, several autoimmune diseases such as systemic lupus erythematosus, myasthenia gravis and rheumatoid arthritis are usually accompanied by thymic hyperplasia, and there are several reports on thymic hyperplasia that could be combined with pSS [33,34,35]. Furthermore, thymectomy in patients with pSS did not appear to improve the clinical symptoms or to decrease the serum level of anti-SSA/Ro antigens in the patient, described by Minato et al. [36]. Similarly, Izumi et al. also reported that after thymectomy in two patients with pSS, the serum ANA levels remained increased vs. control levels [37].

We also noticed the decreased frequency of circulating effector memory CD8+ T cells in patients with pSS. Similarly, Narkeviciute et al. also showed that effector memory CD27+CD57− CD8+ T cells were significantly reduced in the peripheral blood of pSS patients [29]. According to some studies, EM CD8+ T cells were able to migrate to inflamed sites located in peripheral tissues during the effector phase of immune response and took part in pathogens clearance [38]. Furthermore, we found that EM CD8+ T cells were enriched with potentially highly cytotoxic EM3 cells, while EM1 cells were decreased. It is known that EM1 (CD27+CD28+) and EM4 (CD27−CD28+) T cells express low levels of effector mediators such as granzyme B and perforin and high levels of CD127 [16] and are closely related to ‘non-effector’ CM CD8+ cells. Conversely, EM2 (CD27+CD28−) and EM3 (CD27−CD28−) CD8+ T cells express numerous markers of mature effector cells, whereby EM3 cells display stronger ex vivo cytolytic activity and effector cytokine production, resembling TEMRA CD8+ cells. Thus, the EM CD8+ T cell subset in patients with pSS was enriched with effector cells that had high cytolytic activity and were more efficient in migrating to inflamed peripheral tissues than other EM subsets.

Next, we also found that the most mature CD8+ T cells—TEMRA cells—were also altered. We have shown that the frequencies of pre-effector type 1 and pre-effector type 2 cells were decreased, while the levels of CD27+CD28− effector cells were increased in peripheral blood samples from patients with pSS. Previously, it was shown by Sudzius et al. that the levels of effector CD8+ T cells were increased patients with pSS [30]. Conversely, Narkeviciute et al. found that the percentages of effector CD27−CD57−/+CD45RA+CD8+ T cells in pSS patients were lower than in healthy subjects [29]. Therefore, the imbalance of EM and TEMRA CD8+ T cells could be closely linked with the efficient generation of the effector cell subset peripheral lymphoid tissues as well as the migration of mature effector cell to the sites of autoimmune inflammation. Moreover, in an experimental mouse model of pSS (nonobese diabetic mouse model) lacrimal gland infiltrating CD8+ T cells showed an effector cytotoxic phenotype and took part in epithelial cell damage [7]. Our findings also support the idea that CD8+ T cells could be actively involved in the development of pSS.

Finally, we identified four main ‘polarized’ CD8+ T cells in peripheral blood samples from patients with pSS, and we found an imbalance in Tc1, Tc2, Tc17 and Tc17.1 cells in our patients if compared to healthy controls. Primarily, we noticed a decreased relative level of Tc1 cell and found negative correlations of Tc1 subsets with IgG, RF levels, Shirmer test and unstimulated saliva flow. The reduced amount of Tc in the peripheral blood of patients with pSS can be explained by the migration of effector cells to the foci of autoimmune inflammation. It was shown that CXCR3+CCR6− Tc1 cells were capable of killing target cells by secretory degranulation of cytotoxic molecules—granzymes and perforin—releasing as well as effector cytokines secretion—IFNγ and TNFα—which accelerated the tissue inflammation [39]. Interestingly, Jing Zhou et al. showed that administration of a neutralizing anti-TNFα antibody to female NOD mice during the stage prior to disease onset significantly improved salivary secretion, indicating a remission of clinical symptoms of pSS [40]. However, in a randomized controlled trial, Etanercept was an ineffective therapeutic agent in pSS consistent with the absence of suppression of TNFα and other indicators of immune activation in this patient population [41]. Also, increased levels of CXCR3 ligands—CXCL9, CXCL-10 andCXCL-11—were detected in tears of patients with pSS compared with those of patients with non-Sjögren’s dry eye [42]. Furthermore, it was found that Gal1-null mutant mice had high expression of CXCL9 and CXCL10 chemokines in submandibular glands as well as CXCR3+ CD8+ T cells that effectively infiltrated gland tissues if compared with wild-type mice [43]. All the mentioned findings point to the important part of CXCR3-expressing CD8+ T cells in progressive damage to targeting tissues of drying in pSS; as well, targeting CXCR3+ CD8+ T cells may be a new strategy for disease immunotherapy.

Moreover, Tc2 are known for their production of type 2 cytokines (IL-4, IL-5 and IL-13) in conjunction with diminished production of IFNγ and low cytolytic activity, while IL-17-producing CD8+ Tc17 cells are a distinct subset of T cells known for the high production of IL-17A [44]. Our data indicated that in circulating blood from patients with pSS, the level of cytotoxic Tc1 CD8+ T cells that were able to kill infected cells and to secrete effector cytokines (IFNγ and TNFα) was decreased, while the frequencies of regulatory cytokine-producing Tc2 and Tc17 CD8+ T cells were significantly elevated. We found positive correlations of Tc2, Tc17 and Tc17.1 subsets with IgG, RF levels, Shirmer test and unstimulated saliva flow. Little is known about the cellular characteristics and functional activity of Tc2 and Tc17 CD8+ T cells in autoimmune disorders, but, for instance, in patients with rheumatoid arthritis, Tc2 cells were significantly higher than in healthy controls [45]. Similarly, IL-17-producing Tc17 cells are detectible in multiple sclerosis lesions, and patients with early-stage multiple sclerosis had more elevated numbers of Tc17 cells in the cerebrospinal fluid than in peripheral blood [46]. Also, X. Zhang showed that in a murine desiccating stress mouse model, which resembles pSS, CD8+ cell depletion augmented pathogenic Th17 cell generation and consequently worsened the IL-17A-induced disruption of corneal barrier function [11]. However, we were the group that determined these ‘polarized’ CD8+ T cell subsets in patients with pSS, which needs further exploration. We also assumed that imbalance in ‘polarized’ CD8+ T cell subsets may be involved in the pathogenesis of pSS regulating immune functions of cytolytic and cytokine-producing subsets. Herein, we show for the first time that a subset of CD8+ T regulatory cells can significantly mitigate this disease.

## 5. Conclusions

Collectively, our study suggested that CD8+ T cell immune dysfunction plays an im-portant role in the pathogenesis of pSS. Subsets of Tc 1, Tc2 and Tc17 play a role in salivary and lacrimal gland damage, which is confirmed by links with the degree of keratocon-junctivitissicca and xerostomia. It is still unclear to what extent CD8+ T cells and their dis-tinct cell subsetscontribute to the pathology and progression of pSS. Furthermore, the in-volvement of CD8+ T cells in disease initiation and progression remains to be unclear.

## Figures and Tables

**Figure 1 biomedicines-11-02778-f001:**
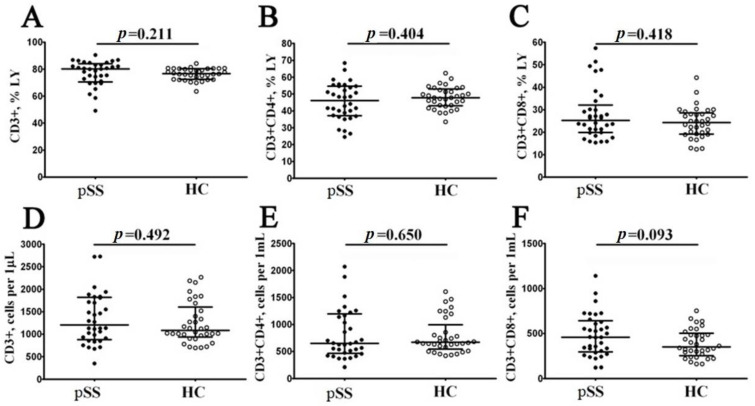
Comparison of relative numbers and concentrations of the main circulating T cell subsets in patients with pSS. The scatter plots (**A**–**C**) and (**D**–**F**) illustrate the percentages (proportion of T cell subset within the total lymphocyte subset) and absolute numbers (count of cells per 1 μL of total peripheral blood) of T cells (CD3+), T-helper cells (Th, CD3+CD4+) and CD8+ T cells (Tcyt, CD3+CD8+), respectively. Patients with Sjögren’s syndrome are represented by black circles (pSS, n = 34), while white circles denote healthy controls (HC, n = 34). Each data point corresponds to an individual subject, and horizontal bars depict the group medians and interquartile ranges (median (Q25; Q75)). Statistical analysis was performed using the Mann–Whitney U test.

**Figure 2 biomedicines-11-02778-f002:**
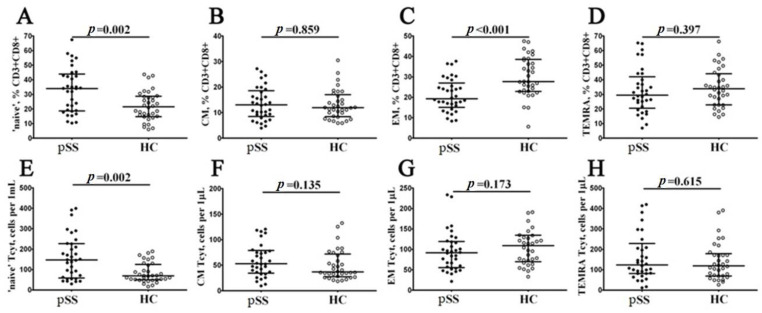
Alteration in relative and absolute numbers of peripheral blood maturation CD8+ T cell subsets in patients with pSS. Scatter plots (**A**–**D**) and (**E**–**H**) show the percentages and absolute numbers of ‘naïve’ (CD45RA+CD62L+), central memory (CM, CD45RA−CD62L+), effector memory (EM, CD45RA−CD62L−) and terminally differentiated CD45RA-positive effector memory (TEMRA, CD45RA+CD62L−) CD8+ T cells, respectively. Black circles denote patients with Sjögren’s syndrome (pSS, n = 34); white circles denote healthy control (HC, n = 34). Each dot represents individual subjects, and horizontal bars depict the group medians and quartile ranges (med (Q25; Q75)). The statistical analysis was performed with the Mann–Whitney U test.

**Figure 3 biomedicines-11-02778-f003:**
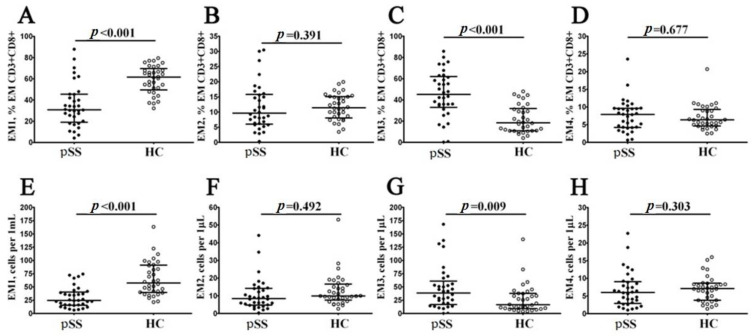
Changes in the relative and absolute quantities of different subsets of EM CD8+ T cells, distinguished by their patterns of CD27 and CD28 expression, were observed in patients with pSS. The data is visually presented using scatter plots, with panels (**A**–**D**) showing the breakdown of EM CD8+ T cells into subsets: EM1 (CD27+CD28+), EM2 (CD27+CD28−), EM3 (CD27−CD28−) and EM4 (CD27−CD28+), expressed in relative values. Another set of scatter plots (**E**–**H**) illustrates the concentrations of EM1, EM2, EM3 and EM4 cells (measured as the number of cells per 1 μL of peripheral whole blood). In these plots, individuals with Sjögren’s syndrome (pSS, n = 34) are represented by black circles, while healthy controls (HC, n = 34) are depicted by white circles. Each individual is denoted by a single dot, and the central horizontal lines indicate the median values of each group, along with the interquartile ranges (med (Q25; Q75)). The statistical analysis was performed using the Mann–Whitney U test.

**Figure 4 biomedicines-11-02778-f004:**
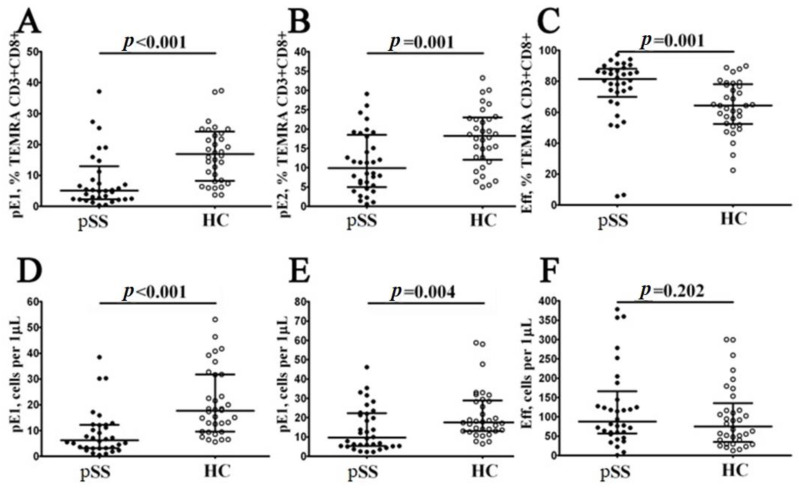
Alterations in relative and absolute number of TEMRA CD8+ T cell subsets with different patterns of CD27 and CD28 expression in patients with pSS. Scatter plots (**A**–**C**) and (**D**–**F**): TEMRA CD8+ T cells were subdivided into CD27+CD28+ pE1, CD27+CD28− pE2 and CD27−CD28− E subsets, respectively. The scatter plots in (**A**–**C**) illustrate the relative proportions of pE1, pE2 and effector cells within the TEMRA CD8+ T cell subset. Meanwhile, the scatter plots in (**D**–**F**) represent the concentrations of pE1, pE2 and effector cells (measured as the number of cells per 1 μL of peripheral whole blood). In these plots, patients with pSS are represented by black circles (n = 34), and healthy controls are depicted by white circles (n = 34). Each data point corresponds to an individual subject, while the horizontal bars show the medians and interquartile ranges (med (Q25; Q75)) for each group. The statistical analysis was performed using the Mann–Whitney U test.

**Figure 5 biomedicines-11-02778-f005:**
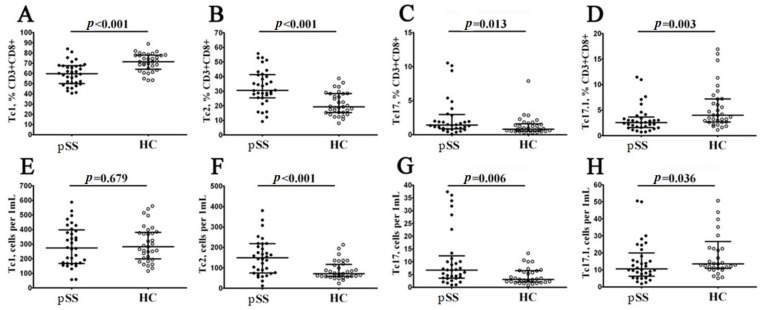
Imbalance in relative and absolute numbers of peripheral blood Tc1, Tc2, Tc17 and double-positive Tc17.1 cells in patients with pSS. Scatter plots (**A**–**D**) and (**E**–**H**): Tc1 (CCR6−CXCR3+), Tc2 (CCR6−CXCR3−), Tc17 (CCR6+CXCR3−) and double-positive Tc17.1 (CCR6+CXCR3+) frequencies, respectively. Scatter plots (**A**–**D**): the relative numbers of Tc1, Tc2, Tc17 and Tc17.1 within total CD8+ T cell subset. Scatter plots (**E**–**H**): Tc1, Tc2, Tc17 and Tc17.1 concentrations (number of cells per 1 μL of peripheral whole blood). Black circles denote patients with Sjögren’s syndrome (pSS, n = 34); white circles denote healthy control (HC, n = 34). Each data point represents individual subjects, and horizontal bars depict the group medians and quartile ranges (med (Q25; Q75)). The statistical analysis was performed using the Mann–Whitney U test.

**Figure 6 biomedicines-11-02778-f006:**
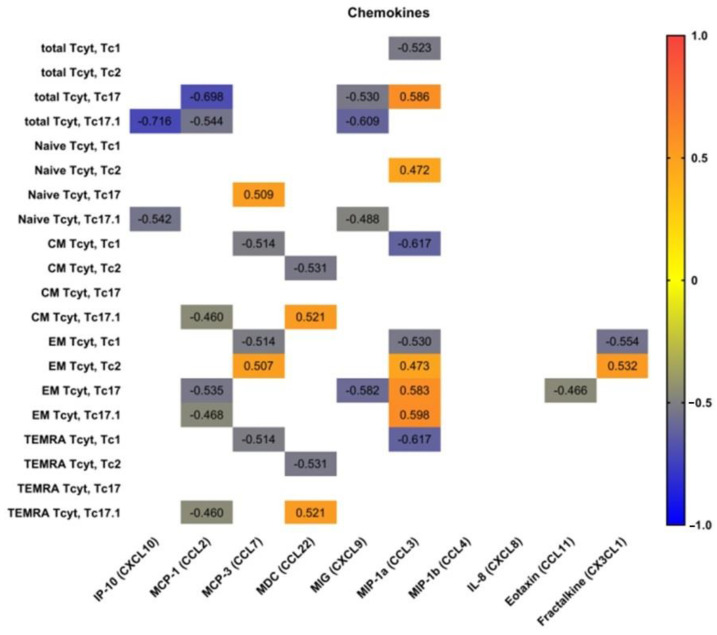
Heat map of correlations between chemokine levels and frequencies of ‘polarized’ CD8+ T cell subsets in patients with pSS (n = 20). Color scale bar shows a range of correlation coefficients (r). Correlation analysis was performed using nonparametric Spearman rank test; only significant correlations are shown (significance was set at *p* < 0.05). The red color signifies a strong positive correlation, gradually transitioning to the blue color on the bar, which indicates a negative correlation.

**Figure 7 biomedicines-11-02778-f007:**
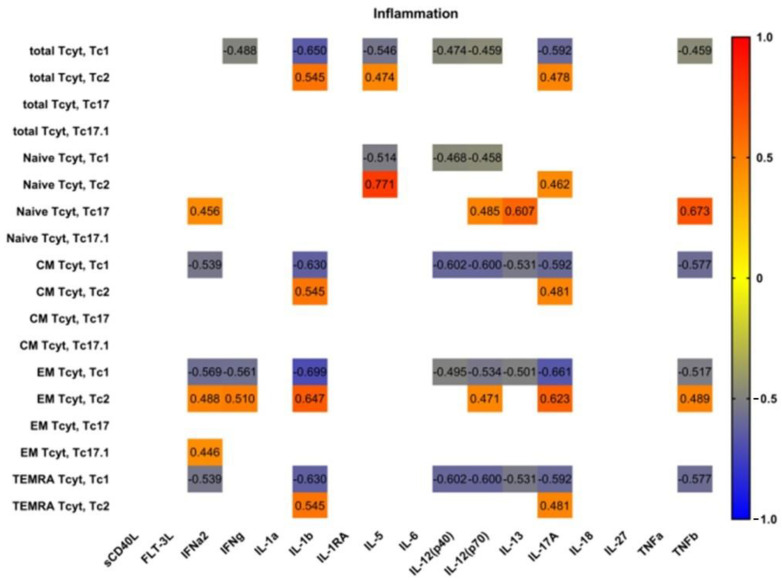
Heat map of correlations between main proinflammatory and effector cytokine levels and frequencies of ‘polarized’ CD8+ T cell subsets in patients with pSS (n = 20). Color scale bar shows a range of correlation coefficients (r). Correlation analysis was performed using nonparametric Spearman rank test; only significant correlations are shown (significance was set at *p* < 0.05). The red color signifies a strong positive correlation, gradually transitioning to the blue color on the bar, which indicates a negative correlation.

**Table 1 biomedicines-11-02778-t001:** Clinical characteristics of pSS patients in this study.

	Sjögren’s Syndrome (n = 34)	Control Subjects (n = 34)
Male/female ratio	2/32	3/31
Age, years, median (IQR)	46 [41; 55]	44 [39; 56]
Disease duration (years), median (IQR)	8 [2.5; 11]	
ESSDAI, median (IQR)	5 [2; 9]	
ESSPRI, median (IQR)	6.7 [5.6; 7.3]	
Clinical manifestations/ESSDAI domain (percentage)		
Constitutional domain	5 (14.7)	
Lymphadenopathy and lymphoma domain	6 (17.7)	
Glandular domain	5 (14.7)	
Articular domain	17 (50)	
Cutaneous domain	5 (14.7)	
Pulmonary domain	2 (5.8)	
Renal domain	0	
Muscular domain	2 (5.8)	
Peripheral nervous system domain	4 (11.6)	
Central nervous system domain	0	
Haematological domain	14 (41.2)	
Biological domain Schirmer test, mm/5 min., median (IQR) Unstimulated sialometry, mL/15 min., median (IQR)	3 (9.7) 3 [1; 9] 1.5 [1; 3]	
Laboratory manifestations (percentage)		
Complement C3 levels below the lower limit of normal value	6 (17.6)	
Complement C4 levels below the lower limit of normal value	3 (8.7)	
Anti-DNA Ab	4 (11.8)	
Rheumathoid factor	10 (29.4)	
Ro-52 Ab	20 (58.8)	
SSA Ab	25 (73.5)	
SSB Ab	10 (29.4)	
CENT B Ab	2 (5.8)	
Medications	28 (82.4)	
Prednisone dose, mg/day, median (IQR)	3.75 [0.0; 10.0]	
Hydroxychloroquine (percentage)	23 (67.7)	
Azathioprine (percentage)	1 (2.9)	
Cyclosporine (percentage)	1 (2.9)	
Mycophenolate mofetil (percentage)	1 (2.9)	
Leflunomide (percentage)	2 (5.8)	
Methotrexate (percentage)	5 (14.7)	

IQR—the interquartile range; ESSDAI—Sjögren’s Syndrome Disease Activity Index; ESSPRI—Sjögren’s Syndrome Patient Reported Index; Anti-DNA—double-stranded DNA, Ab—antibodies.

## Data Availability

Not applicable.

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
