# Peer review of "Circulating CD8+ T Cell Subsets in Primary Sjögren’s Syndrome"

_biomedicines, 2023, doi:10.3390/biomedicines11102778_

Round 1
Reviewer 1 Report
With interest, I read the manuscript biomedicines-2595412. I have only one major and a few minor comments only.
Major comment:
1. Comparative characteristics of the controls needs to be added to Table 1, at least basic demographics.
Minor comments:
1. Abbreviations used in Table 1 must be explained in its legend.
2. It should be “6. Conclusions” not “5. Conclusions”.
3. “Alexey Maslyanskiy 1 and 4 †”. What do you mean “and”? Are there any further authors?
4. Figures 6 and 7 are to small to read their details.
Minor-moderate editing of English language required
Reviewer 2 Report
In this article, Kudryavtsev et al assessed the imbalance of subsets within CD8+ cells in peripheral blood. They analzyed blood samples from 34 patients with pSS and 34 healthy individuals as controls by flow cytometry. They revealed that both the relative and absolute numbers of naïve CD8+ T cells were higher in pSS patients compared to the healthy controls. The proportions of effector memory CD8+ T cells were notably lower. They also suggested that among patients with pSS, the levels of cytotoxic Tc1 CD8+ T cells were reduced, while the frequencies of regulatory cytokine-producing Tc2 and Tc17 CD8+ T cells were significantly elevated. The Tc1 cell subsets displayed a negative correlation with immunoglobulin G, Rheumatoid factor, the Schirmer test, and unstimulated saliva flow. The Tc2 cell subsets exhibited a positive correlation with these parameters. They concluded that immune dysfunction within CD8+ T cells, including alterations in Tc1 cells, plays a significant role in the development of pSS. It has been examined in detail with flow cytometry and can be evaluated to a certain degree. I have some questions as follows.
Major concerns)
1) The correlation between CD8+ subsets and antibodies, etc. is also discussed. In Discussion, the relationship between Tc2, Tc17, etc. and B cells should also be discussed. Recently, it has been reported that B cells interact with T cells and each other through cytokines (Fukasawa T, Yoshizaki A, Ebata S, Yoshizaki-Ogawa A, Asano Y, Enomoto A, Miyagawa K, Kazoe Y, Mawatari K, Kitamori T, Sato S. Single-cell-level protein analysis revealing the roles of autoantigen-reactive B lymphocytes in autoimmune disease and the murine model. Elife. 2021 Dec 2;10:e67209. doi: 10.7554/eLife.67209. PMID: 34854378; PMCID: PMC8639144.).
It is assumed that CD8+ cells also interacts with B cells, since some T cells produce IL-4, 5, 13, 17, etc. Citing the paper above, like the interaction of CD4+ T cells to B cells and the interaction of B cells to CD4+ T cells, please discuss the interaction from CD8+ cells to B cells and the interaction from B cells to CD8+ cells.
Minor concerns)
1) In line 444, you wrote "Similarly, Izumi et al. also reported that after thymectomy in two patients with pSS the serum ANA levels remained increased vs. control levels [37]." Please delete additional "space" between "also" and "reported."
2) In 485, you wrote "administration of a neutralizing anti-TNF-α antibody to female NOD mice during the stage prior to disease onset significantly improved salivary secretion, indicating a remission of clinical symptoms of SS [41]. Нowever, in Randomized Controlled Trial Etanercept was an ineffective therapeutic agent in pSS consistent with the absence of suppression of TNFalpha and other indicators of immune activation in this patient population."
The font is wrong in this section, please correct it.
3) In line 569, you wrote "12. : Seror R, Bowman SJ, Brito-Zeron P, et al. EULAR Sjögren’s syndrome disease activity index (ESSDAI): a user guide. RMD Open 2015;1:e000022"
Please delete ":".
Round 2
Reviewer 2 Report
The authors responded my questions adequately. No additional comments.